# Individual and Group-Based Effects of In Vitro Fiber Interventions on the Fecal Microbiota

**DOI:** 10.3390/microorganisms11082001

**Published:** 2023-08-03

**Authors:** Valeria Agamennone, Tim J. van den Broek, Alie de Kat Angelino-Bart, Femke P. M. Hoevenaars, Jan Willem van der Kamp, Frank H. J. Schuren

**Affiliations:** Microbiology and Systems Biology Group, TNO, 2333 BE Leiden, The Netherlands

**Keywords:** dietary fiber, in vitro model, individual microbiota, inflammatory bowel disease, SCFA

## Abstract

The development of microbiome-targeted strategies is limited by individual differences in gut microbiome composition and metabolic responses to interventions. In vitro models that can replicate this variation allow us to conduct pre-clinical studies and assess efficacy. This study describes the exposure of 16 individual fecal microbiota samples to 5 different fibers using an in vitro system for the anaerobic cultivation of bacteria. The individual microbiota differed in composition and metabolite profiles (short-chain fatty acids and branched-chain fatty acids) after incubation with the fibers. Furthermore, microbiota composition after fiber incubation was significantly different between subjects with good intestinal health and subjects with Inflammatory Bowel Disease (IBD). α-diversity was differently affected by dietary fibers; for example, exposure to psyllium resulted in increased diversity in the healthy group and in decreased diversity in the IBD group. Instead, the functional metabolic profile did not differ between the two groups. Finally, the combination of all fibers, tested on the microbiota from IBD subjects, resulted in stronger overall effects on both microbiota composition and metabolite production compared to the single fibers. These results confirm that incubation with dietary fiber results in different compositional and functional effects on individual microbiota and that in vitro models represent successful tools for studying individual fiber effects.

## 1. Introduction

A healthy diet is crucial to maintain, promote, and restore health. Although some dietary patterns are known to positively influence health and reduce the risk of non-communicable diseases [1], a one-size-fits-all approach in nutrition is not always accurate: people differ in their genetic background, lifestyle habits, and microbiota, so they might show different responses to a particular dietary approach [2]. Therefore, within the general guidelines informed by clinical and epidemiological data, there is space for personalized advice aimed at promoting health [3]. This approach depends on our understanding of the many different levels of inter-individual differences (health biomarkers, lifestyle, and microbiome) and the ways that they can influence the outcomes of interventions. Models and tools can help break down the complexity of the system and understand how different factors influence it [4].

Fibers are dietary components that are non-digestible by human digestive enzymes and associated with positive intestinal and systemic health effects. Higher fiber intake is linked to lower body weight and decreased risk for certain conditions, including obesity, diabetes, cardiovascular, and gastrointestinal diseases [5]. To a great extent, the health benefits of fibers are mediated by their effect on the gut microbiota [6]. In this regard, an important mechanism of action is fiber fermentation: in the colon, bacteria can break down non-digestible carbohydrates, and the fermentation products, such as short-chain fatty acids (SCFA), provide energy to colonic epithelial cells. SCFA can also enter the bloodstream, thereby reaching and providing energy to other parts of the body. Fermentation and the production of SCFA also influence the pH in the colon, contributing to maintaining homeostasis and protecting from invading pathogens.

Each person has a unique gut microbiota, potentially resulting in different metabolic responses and health outcomes to the same food ingredient [7]. Indeed, there is evidence that enterotypes can influence the outcome of dietary interventions [8]. For fibers specifically, this happens because the microbiota of different individuals may ferment dietary fibers with different efficiencies [9,10]. Dysbiosis, the alteration in microbiota composition occurring in digestive diseases, can also hinder the beneficial effects of fiber interventions. Therefore, while dietary fiber, in general, is known to positively influence health, the question is whether different fibers would better suit different individuals and whether people, including those affected by diseases of the digestive tract, would benefit from personalized fiber recommendations. A review from Armstrong et al. highlights the need to conduct more research on the effects of fiber in dysbiotic settings, such as in IBD, and the importance of considering the effects of different fiber types [11].

This study describes the exposure of 16 individual fecal microbiota samples to 5 different fibers using an in vitro system for the anaerobic cultivation of bacteria [12]. After fiber exposure, effects on microbial composition and functional profiles were determined. Furthermore, we investigated whether the intestinal health status of fecal donors modulates fiber effects on the microbiota. To this aim, we evaluated the fecal microbiota obtained from donors with good intestinal health and from donors with a diagnosis of IBD.

## 2. Materials and Methods

### 2.1. Fecal Collection

Figure 1 illustrates the procedures performed during the study. Fecal material for the experiment was collected from 16 donors, including 5 with good intestinal health (3 males and 2 females; age between 17 and 45; average age = 32) and 11 with intestinal complaints and a clinical diagnosis of Inflammatory Bowel Disease (IBD) (8 males and 3 females; age between 32 and 74; average age = 52). For simplicity, the two groups of subjects are referred to as ‘healthy’ and ‘IBD’ throughout the article; these labels should be interpreted as descriptors of their intestinal, and not systemic, health. The fecal donors were recruited by advertising the research project at a university hospital in The Netherlands. The collection of fecal samples was approved by the Medical Research Ethics Committee (METC) of the University Medical Center Utrecht, an independent ethical review board (Central Committee on Research Involving Human Subjects (CCMO) number: NL62046.041.17).

To collect the sample, the subjects used the FecesCatcher, a specimen collection device consisting of biodegradable paper to be placed under the toilet seat (fecesvanger.nl). Fecal material was collected with a sterile plastic spoon and placed in a tube that was, in turn, placed in an anaerobic jar (materials provided by TNO) with an AnaeroGen Sachet (Thermo Fisher Diagnostics GMBH). The jar was kept at 4 °C until delivery at the laboratory (within 24 h from collection). There, the material was introduced in an anaerobic chamber, diluted 1:3 with phosphate-buffered saline, and homogenized. Finally, 20% glycerol was added before storing the material at −80 °C.

### 2.2. Fiber Selection and Preparation

Five fibers with different composition, structure, and solubility were selected: β-glucan (Oat β-glucan High viscosity, product code P-BGOH, CAS number 9041-22-9, Bio-connect, Megazyme, Bray, Ireland), resistant starch (HI-MAIZE^®^ 260 resistant starch, product code 22000B00, CAS number 9005-25-8, Ingredion, Westchester, IL, USA), psyllium (VITACEL^®^ Psyllium P 95, J. Rettenmaier & Söhne GmbH, Rosenberg, Germany), cellulose (50% VITACEL oat Hull Fiber HF 101 and 50% VITACEL oat Hull Fiber HF 600, Rettenmaier & Söhne GmbH), and pectin (50% Lemon Pectin Degree of Methylation DM30 and 50% Lemon Pectin DM67, Prof. Dr. H.A. Schols, Wageningen University).

A stock solution was prepared by weighing 80 mg of fiber and adding 10 mL of Milli-Q water. The stock solution was incubated overnight at 1200 rpm at 25 °C to completely dissolve the fiber, and it was further diluted 2× to achieve the test concentration (4 mg/mL). Additionally, all fibers were combined in equal proportions to create a fiber mix that was tested on fecal material from IBD donors. The fiber mix was tested at two concentrations: 4 mg/mL, for comparison with the single fibers, and 12 mg/mL, representative of a high-fiber intervention.

### 2.3. Anaerobic Incubations

The fecal material was incubated anaerobically in the i-screen (intestinal screening) system [12]. First, the fecal samples were pre-cultured overnight in modified standard ileal efflux medium (SIEM) in anaerobic conditions, at 37 °C, and with shaking at 300 rpm [13]. The microbiota was then transferred to microtiter plates, and the fibers were added at a concentration of 4 mg/mL. The i-screen incubation started with a fecal bacterial load of approximately 10^9^ CFU/mL. The microbiota was cultured in SIEM with pH adjusted to 5.8. All compounds were tested in triplicate. After 24 h of anaerobic fermentation, the incubation material was sampled for DNA isolation and metabolite analysis.

### 2.4. DNA Isolation

Following incubation, samples were collected, and DNA was isolated as described by Ladirat et al. [13] with minor adjustments: samples were mixed with 300 μL lysis buffer (Agowa, Berlin, Germany), 500 μL zirconium beads (0.1 mm), 500 μL phenol, and bead-beaten for 3 min in a Bead Beater (BioSpec Products, Bartlesville, OK, USA).

### 2.5. Amplicon Sequencing

Changes in the microbiota composition were analyzed by using 16S rDNA amplicon sequencing. The V4 hypervariable region was targeted. A total of 100 pg of DNA was amplified as described by Kozich et al. [14], with the exception that 30 cycles were used instead of 35, applying F515/R806 primers [15]. Primers included Illumina adapters and a unique 8 nt sample index sequence key [14]. The amplicon libraries were pooled in equimolar amounts and purified using the QIAquick Gel Extraction Kit (QIAGEN, Hilden, Germany). Amplicon quality and size were analyzed on a Fragment Analyzer (Advanced Analytical Technologies, Inc., Heidelberg, Germany). Paired-end sequencing of amplicons (approximately 400 base pairs) was conducted on the Illumina MiSeq platform (Illumina, Eindhoven, The Netherlands).

Sequence pre-processing, analysis, and classification were performed using modules implemented in the Mothur software platform [16]. Chimeric sequences were identified and removed using the chimera.uchime command. 16S rDNA unique sequences were aligned using the ‘align.seqs’ command and the Mothur-compatible Bacterial SILVA SEED database (Release 119). Taxonomic classification was performed using the RDP-II Naïve Bayesian Classifier using a 60% confidence threshold against the RDP Database (Release 11.1) for 16S rRNA. Taxonomic classification was performed at the genus level.

### 2.6. Metabolite Analysis

SCFA covering acetate, propionate, and n-butyrate and branched chain fatty acids (BCFA) covering iso-butyrate and iso-valerate were analyzed as described by Jouany [17], with modifications as described by Van Nuenen [18]. Briefly, exposed material from the i-screen samples was centrifuged (~12,000× *g*, 5 min). Clear supernatant was filter-sterilized (0.45 µm). A mixture of formic acid (20%), methanol, and 2-ethyl butyric acid (internal standard, 2 mg/mL in methanol) was added. A 3 µL sample with a split ratio of 75.0 was injected on a GC-column (ZB-5HT inferno, ID 0.52 mm, film thickness 0.10 µm; Zebron; Phenomenex, Torrance, CA, USA) in a gas chromatograph (GC-2014, Shimadzu Kyoto, Japan).

### 2.7. Data Analysis—16S

All data analysis was performed using R version 4.1.2 [19]. All figures were composed using the ggplot2 package version 3.3.5 [20].

Univariate regression on the microbiome count data was in all cases performed using linear mixed models from the dream package, version 1.24.0, with the variancePartition extension [21,22]. This combination of packages provides functions for differential abundance testing using negative binomial linear mixed models for repeated measures data. In all models, ‘subject’ was taken as a random factor.

Multivariate analysis and ordinations were performed using the vegan package, version 2.5-7 [23]. This package was also used to calculate the inverse Simpson and Shannon alpha-diversities.

The multivariate models fitted by PERMANOVA were tested via permutation analysis in order to produce Type III (marginal) *p*-values for the terms included in the model. A total of 103 permutations were used for all reported results. Count data were normalized using the Wisconsin double standardization after square-root transformation in the case of PERMANOVA, RDA, and MDS ordinations. In the case of the distance analysis, PERMANOVA, and MDS, the Bray–Curtis distance measure was used.

The 16S data were filtered to include only those taxa that contribute to the first 95% of all counts in the data. This step was not performed for the alpha-diversity analysis. The filtering procedure consisted of the steps denoted below.

Given the count table of taxa A, where A is an m, n matrix (samples by taxa), let the row normalized matrix be denoted by D, where D is an m, n matrix, with the formula given by
$$D_{i,j}=\frac{A_{i,j}}{\sum_{\theta =1}^{n}\;A_{i,\theta }}$$

Let the row count normalized column sums of our matrix D be denoted as the vector $\hat{c}$, given by
$${\hat{c}}_{j}=\sum_{\theta =1}^{m}\;D_{\theta ,j}\forall j\in [1,n]$$

The cumulative sum of the column sums normalized for column count is then denoted as C, where
$$C=\sum_{\theta =1}^{n}\;\frac{{\hat{c}}_{\theta }}{n}$$

Any taxa where C is smaller than 0.95 (corresponding to 95%) was then chosen to be included in the analysis. This procedure eliminates sparse, low-count taxa from the dataset.

### 2.8. Statistics

Statistical analysis of the SCFAs and α-diversity measures was performed using the lme4 and lmerTest packages, with the emmeans package for post hoc analysis [20,21,22]. All models use ‘subject’ as a random effect.

## 3. Results

### 3.1. Fiber’s Effects on α-Diversity

The overall effect of fiber on the α-diversity was significant for the healthy microbiota (*p* < 0.001 for overall fiber effect) but not for the IBD microbiota (*p* = 0.0501). Linear model analyses indicate a significant group-treatment effect (*p* < 0.001), meaning that the effect of fiber on microbiota diversity was significantly different between the healthy and the IBD groups. Figure 2 shows more specifically which fibers had a significant effect on the diversity in each of the two groups: in the healthy group (Figure 2a), α-diversity was significantly decreased after treatment with pectin compared to the untreated control (*p* < 0.001), and it was significantly increased after treatment with psyllium (*p* = 0.015) and resistant starch (*p* = 0.022); in the IBD group, α-diversity was significantly decreased after treatment with psyllium compared to the untreated control (*p* = 0.016).

Within each group, fibers had a subject-specific effect on α-diversity: the microbiota of some donors showed a stronger change in α-diversity compared to the untreated control after exposure to specific fibers (Appendix A).

### 3.2. Fiber’s Effects on Microbiota Composition

After 24 h of fiber incubation, the microbiota composition was significantly different from the untreated control, both in the healthy group (F = 3.17; *p* = 0.001) and in the IBD group (F = 4.39; *p* = 0.001). Furthermore, ANOVA revealed a significant subject–treatment interaction (*p* = 0.002 for the healthy group, *p* = 0.001 for the IBD group), meaning that the effect of fiber on microbiota composition was significantly different across individuals (Appendix A). Still, differences in composition between individual microbiota were larger than differences induced by fiber treatments (healthy group: F = 17.08, *p* = 0.001; IBD group: F = 24.12, *p* = 0.001). This is illustrated in Figure 3: here, the individual microbiota (represented by dots of different colors and shapes) cluster based on donor subjects (color) rather than based on fiber treatment (shapes).

PERMANOVA revealed that, after incubation with fibers, microbiota composition was significantly different between the healthy and the IBD group (*p* = 0.001 for the overall group–treatment interaction effect). Specifically, different bacterial genera were affected by fiber treatment in the healthy versus the IBD microbiota. Figure 4 shows the bacteria whose abundance was significantly (*p* < 0.001) affected by the fibers in the healthy and IBD microbiota, compared to the untreated control. *Lachnospira* was strongly increased in both groups after treatment with pectin. Of all tested fibers, pectin affected the highest number of bacterial taxa in both the healthy and the IBD microbiota; the second largest effect was shown by β-glucan on the IBD microbiota.

In the healthy microbiota, exposure to pectin mostly resulted in decreased abundances of some taxa compared to the untreated control. Other significant effects observed in the healthy group included a decrease in *Escherichia*/*Shigella* after treatment with psyllium and an increase in *Blautia* after treatment with psyllium or β-glucan. Overall, the IBD microbiota showed more significant effects compared to the healthy microbiota. Many bacteria increased in abundance after fiber treatment. Notably, *Faecalibacterium* increased after exposure to pectin, and *Bifidobacterium* increased after exposure to any fiber.

### 3.3. Fiber’s Effects on Metabolite Levels

The overall effect of fiber, both in the healthy and IBD group, was significant on all metabolites measured: acetate, propionate, n-butyrate, iso-valerate, iso-butyrate, valerate, and 2-methylbutyrate. The largest effect of fiber was observed on acetate (healthy group: F = 217, *p* << 0.0001; IBD group: F = 138, *p* << 0.0001). Figure 5 provides more detail, showing metabolite levels in the individual microbiota after incubation with fiber and in untreated control conditions. In the healthy group, acetate and propionate were significantly increased after treatment with pectin, psyllium, and β-glucan compared to the untreated control. In the IBD group, SCFA (acetate, propionate, and n-butyrate) increased after treatment with β-glucan and decreased after treatment with cellulose. Both in the healthy and the IBD group, β-glucan was the fiber causing the highest increase in SCFA, followed by pectin. β-glucan and pectin were the only fibers that significantly increased acetate levels compared to the control in both healthy and IBD microbiota. Appendix A shows the effects of fiber on the overall metabolite levels of individual microbiota compared to the untreated control.

Linear mixed-effect models indicate that fiber’s effects on metabolite levels are significantly different across individual microbiota (*p* < 0.005 for subject-treatment interaction). This effect was observed for all metabolites except acetate. This means that the individual microbiota had different metabolite profiles after exposure to fiber; however, their acetate profile was similar. Interestingly, an opposite trend was observed for the group–treatment interaction: the effect of fiber on metabolite levels was not significantly different between healthy and IBD subjects, except for acetate (*p* < 0.005). This means that the effect of fiber on acetate levels may be mediated by the microbiota composition of the person.

### 3.4. Effects of Single Fibers and Fiber Mix on Microbiota Composition and Metabolites

When comparing fiber-exposed microbiota to untreated microbiota, the fiber mix has an overall larger effect than the single fibers. This is visible in Figure 6, which shows that the fiber-treated microbiota is more different from the untreated control after exposure to fiber mixes than after exposure to single fibers. This applies both in terms of microbiota composition (Figure 6a) and metabolite levels (Figure 6b). Among the single fibers, pectin and β-glucan have the largest effect on the microbiota, and cellulose has the smallest effect on microbiota composition compared to the untreated control.

Canonical correspondence analysis (CCA) illustrates the effect of different fibers on individual bacterial genera in the microbiota (Figure 7). The pink dots represent the microbiota composition associated with the different fibers, and the distance between the dots is proportional to the difference in microbiota composition. Based on the distance between the fiber and the untreated control, the plot confirms that the fiber mixes have the largest effect on microbiota composition compared to the untreated control, whereas cellulose has the smallest effect of all fibers on microbiota (also visible in Figure 6a). The plot also indicates that different abundances of *Prevotella* and *Lachnospira* are what differentiates most between fiber treatments, as the distribution of the fibers along axis CCA2 shows: *Prevotella* increases in concentration from pectin, untreated control, cellulose, psyllium, and resistant starch to β-glucan, while *Lachnospira* increases from β-glucan, resistant starch, psyllium, cellulose, and control to pectin.

## 4. Discussion

This study described the exposure of 16 individual fecal microbiota samples to 5 different fibers in vitro. Effects on microbial composition and functional profiles were determined, and we explored whether these effects differed among individuals and between two groups with different intestinal health.

All fibers tested in this study influenced the microbiota, albeit with individual variations in effect size. However, the differences in microbiota composition between individual study subjects were larger than the differences induced by fiber treatments. The α-diversity was significantly decreased after treatment with pectin, but only in the healthy group. Treatment with psyllium resulted in a significant increase in α-diversity in the healthy group and a significant decrease in α-diversity in the IBD group. High diversity and richness of the gut microbiota are generally considered biomarkers of intestinal health, but there is also evidence that diets rich in fiber, while inducing positive metabolic responses, also reduce overall microbial diversity by promoting the growth of specific bacteria [24,25].

Fibers can produce specific effects on the microbiota depending on their structural properties, solubility, and fermentability [26]. In their model, Cantu-Jungles and Hamaker illustrate how the structural complexity of fibers can contribute to their specificity: fibers with highly complex structures have high specificity because only a small number of gut bacteria have the enzymatic repertoire required to degrade them, whereas fibers with more simple structures are less specific [27]. Fibers that are digested by keystone species also have low specificity because the metabolic products released become accessible to other gut bacteria through cross-feeding [28]. Based on this model, fibers with high specificity are expected to elicit similar effects on the microbiota of different individuals, whereas fibers with low specificity are expected to cause effects that are individual-specific [27]. In this study, we observed that the water-soluble and high-molecular-weight fibers β-glucan and pectin have, when applied at the same level, a bigger effect on the composition and metabolite profile of the microbiota than less fermentable and insoluble fibers such as resistant starch and cellulose. However, it should be noted that insoluble fibers are being used in food products at considerably higher levels than highly viscous soluble fibers.

In this study, pectin had an especially considerable effect, causing significant shifts in many bacterial groups. Interestingly, these microbiome changes were often related to the intestinal health status of the microbiota donor. For example, *Lachnospira* was significantly increased after exposure to pectin in both the healthy and the IBD groups. *Bifidobacterium* and *Faecalibacterium*, two bacterial genera that are considered beneficial for gut health because of their capacity to ferment dietary fiber [29] and to produce butyrate [30], were significantly increased after exposure to pectin, but only in the IBD group. The same was observed after exposure to β-glucan: *Bifidobacterium* and *Faecalibacterium* were only increased in the IBD group. Furthermore, we observed that fiber mixes promote bigger shifts in microbiota composition compared to single fibers: this is easily explained because fiber mixes contain a variety of different molecules.

The mechanisms that modulate the health effects of fibers include the production of specific metabolites, such as short-chain fatty acids (SCFAs). Aside from shifts in microbiota composition, it is, therefore, important to consider the functional effects of fiber interventions. The two can go along hand in hand, and individual differences in microbiome composition can result in different functional responses to dietary fiber. Indeed, the microbiota of different enterotypes has been shown to produce different levels and ratios of SCFA after in vitro fiber fermentation [10]. The microbiota of different individuals, therefore, has different capacities to ferment dietary fibers, and SCFA response can be predicted by microbiome shifts and baseline microbiota [31,32]. Still, because of functional redundancy, the effects of some fibers may be comparable across individuals despite differences in microbiota composition. Here, we show that different fibers lead to comparable acetate levels in the microbiota from different donors (although, interestingly, acetate levels are significantly different between the healthy and the IBD group). We also observe another microbiota-independent effect, namely, that β-glucan promotes SCFA production to a larger extent than the other fibers across individuals. These observations suggest that it may be possible to predict fiber effects on some metabolites regardless of background microbiota and that some fibers may be successfully used to promote intestinal as well as systemic health effects in many subjects. In this study, β-glucan has the largest effect on SCFA levels, and pectin has the largest effect on microbiota composition compared to the untreated control. These findings support the prebiotic potential of these fibers, exerted through microbiome modulation or SCFA production [33]. β-glucan promotes health by decreasing the adsorption of fats, including cholesterol, through the intestinal epithelium [34], and recent findings show that the beneficial effects of this fiber are also modulated by the microbial production of SCFA and bile acid metabolism [35]. Pectin is also known to promote gut health by reducing inflammation through microbial-dependent and -independent pathways [36]. This study highlights the importance of contemplating that the effects of these fibers are modulated by differences in microbiota composition, whether among individuals or between health groups.

The small sample size represents a limitation in this study, creating a risk of bias and the presence of outliers. The healthy group included five subjects, most likely not enough to represent the diversity of a healthy population. Especially in microbiome studies, considering the complexity of gut bacterial populations and the high interindividual diversity, larger sample sizes are needed to detect significant effects and differences between subjects [37]. Still, even in large populations, it has been shown that only a small proportion of the total microbiome diversity can be accounted for by known factors, such as diet, lifestyle habits, health status, and medicine use, with more than 80% of interindividual variation remaining essentially unexplained [38]. This study included more subjects than previous similar in vitro works [26,39], and it included individuals with different intestinal health. Furthermore, different types of fibers were investigated, allowing us to observe that ingredients with different physicochemical properties can result in specific microbial and metabolic changes. Functional changes in response to fibers do not always result in improvements in metabolic health [40]. Therefore, future studies should aim to determine whether these changes also occur in vivo. If so, the individual responses to fiber exposure observed in vitro could be predictive of in vivo responses and could therefore guide the formulation of personalized advice for fiber intake. Naturally, such validation studies should be performed at a larger scale and would benefit from the inclusion of functional analyses (metatranscriptomics, in addition to metabolite measurements) to achieve a better characterization of responses to microbiome interventions [41].

Another limitation of this study was the lack of baseline information that prevented us from studying microbial responses, i.e., differences between microbiota composition and function before and after fiber fermentation. To overcome this issue, we compared taxonomic and SCFA patterns in the fiber-treated samples to the untreated control samples to infer fiber effects. Future studies should include baseline data to determine whether microbiome profiles can be used as predictors of responsiveness or non-responsiveness to fiber interventions [24].

## 5. Conclusions

This study described the in vitro exposure of individual fecal microbiota to different fibers and showed that in vitro models could effectively detect compositional and functional differences between individual microbiota. The effects of dietary fibers observed in this study were significantly different among individual microbiota but also between the healthy and IBD groups. Furthermore, the combination of all fibers resulted in stronger effects on microbiota composition and metabolite production compared to single fibers. It seems in vitro models are well suited to study fibers in early screening stages and can be applied to test hypotheses about the specific effects of novel ingredients or combinations thereof. Future studies should evaluate to what extent observations from in vitro models, such as the i-screen applied here, translate to in vivo situations.

## Figures and Tables

**Figure 1 microorganisms-11-02001-f001:**
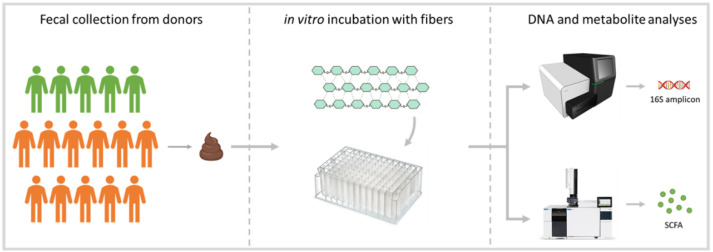
Study design. Fecal material was collected from 16 donors (5 healthy individuals: green; 11 individuals with IBD: orange) and incubated in the in vitro gut model in the presence of a fiber (4 mg/mL) or a fiber mix (4 and 12 mg/mL). After 24 h of incubation, samples were collected for 16S rDNA sequencing and metabolite (short-chain and branched-chain fatty acids) analyses.

**Figure 2 microorganisms-11-02001-f002:**
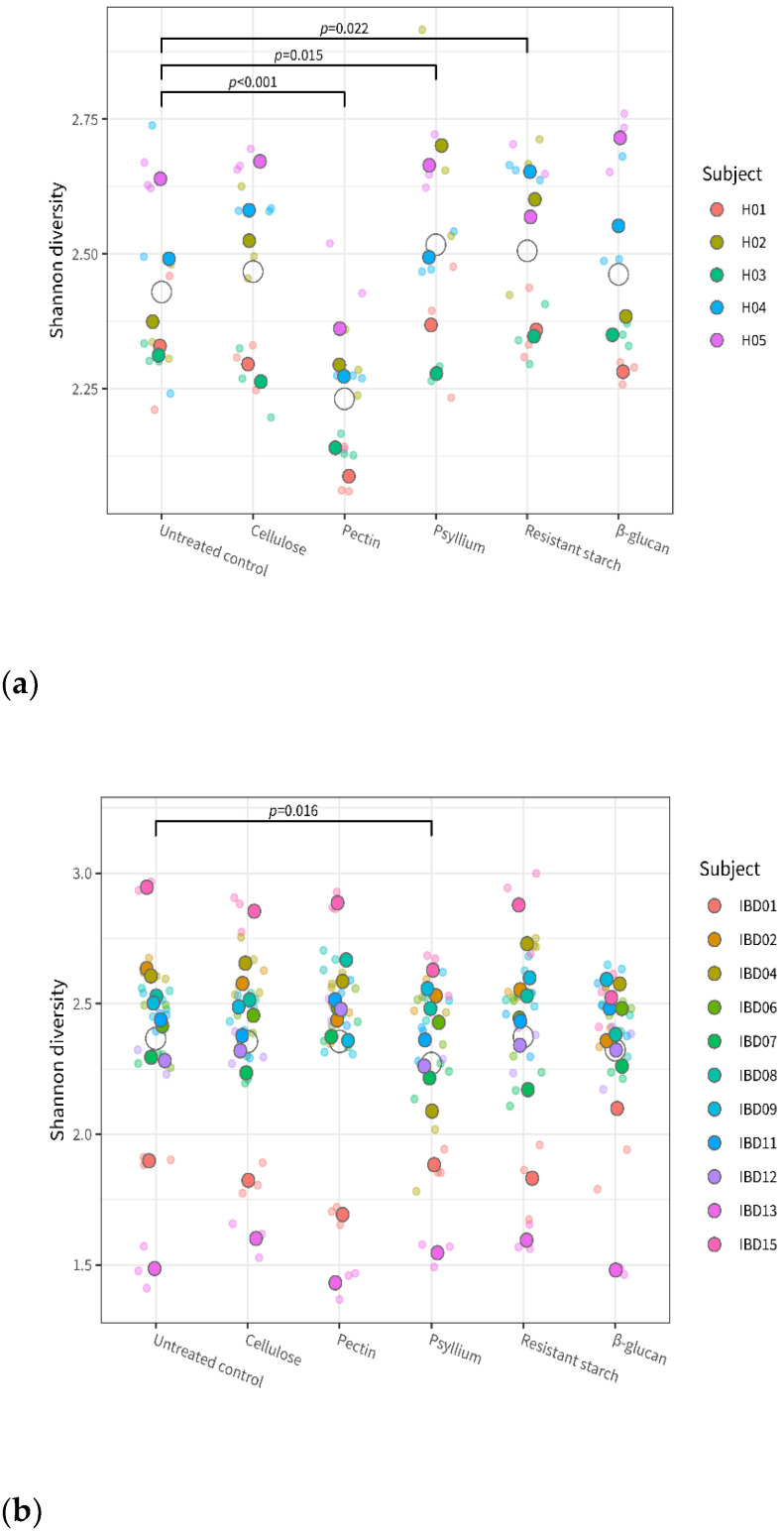
Alpha diversity (Shannon index) of the individual microbiota from healthy (**a**) and IBD (**b**) subjects after treatment with the different fibers and in control conditions. Dots of different colors indicate the individual donors, with smaller and more transparent dots indicating the individual replicates, and larger and darker dots indicating the average of triplicates; open circles represent the average for each test condition. Values are given for each of the fiber treatments and for the untreated control. The given *p*-values indicate significant difference between the corresponding fiber treatment and the control (across all subjects from the same health group).

**Figure 3 microorganisms-11-02001-f003:**
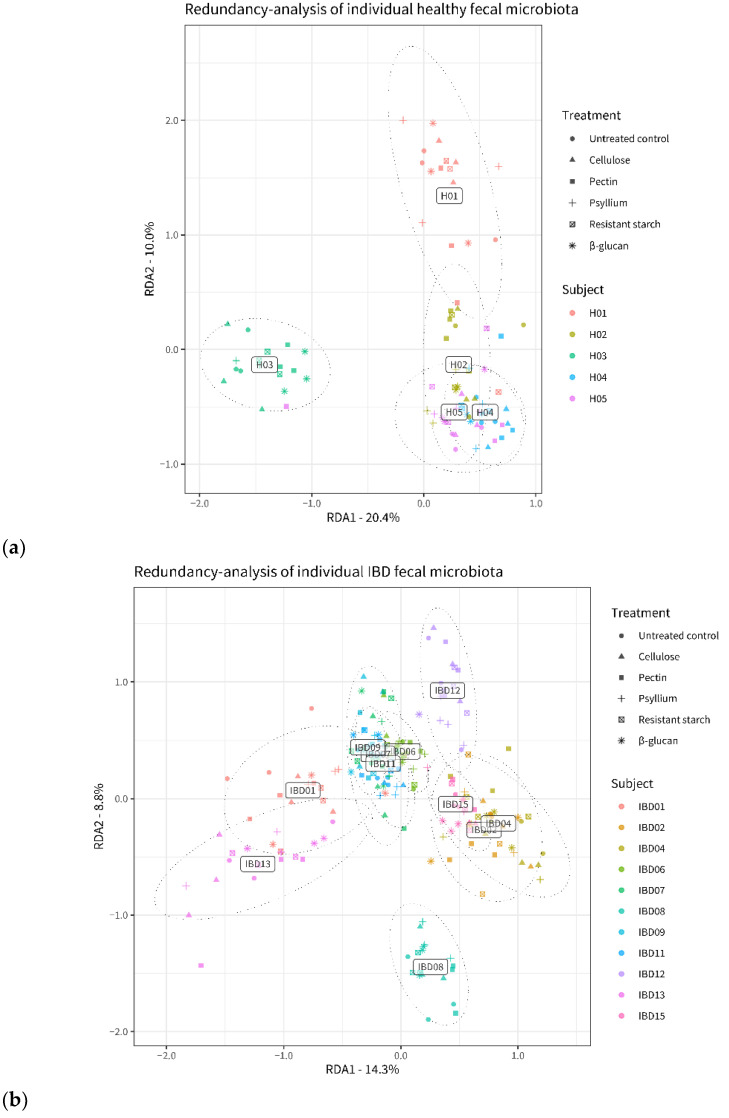
Distances between the microbiota of healthy (**a**) and IBD (**b**) subjects after incubation with the different fibers and in control conditions. The plots are based on Redundancy Analysis (RDA) of center-log-ratio transformed microbiome data and include taxa representing 95% of all classified reads. Dots represent individual microbiota samples (*n* = 3), and their position is indicative of the microbiota composition: more distant dots indicate more dissimilar microbiota samples. Colors indicate the individual donors; shapes indicate the different fibers.

**Figure 4 microorganisms-11-02001-f004:**
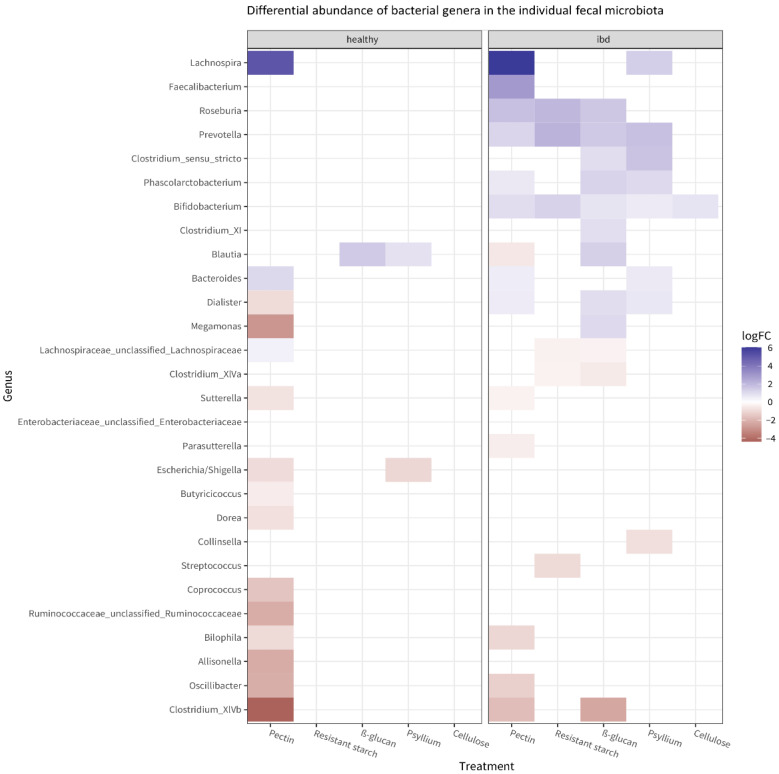
Heatmap of the bacterial genera whose abundance is significantly affected by the fibers (*p* < 0.05) in the microbiota from healthy and IBD subjects. The heatmap shows which bacteria have significantly different abundance after fiber treatment compared to the untreated control, as well as the direction of change (blue: increase compared to untreated control; red: decrease compared to the untreated control). The effects of fiber on the most abundant taxa in each individual microbiota are shown in Appendix A.

**Figure 5 microorganisms-11-02001-f005:**
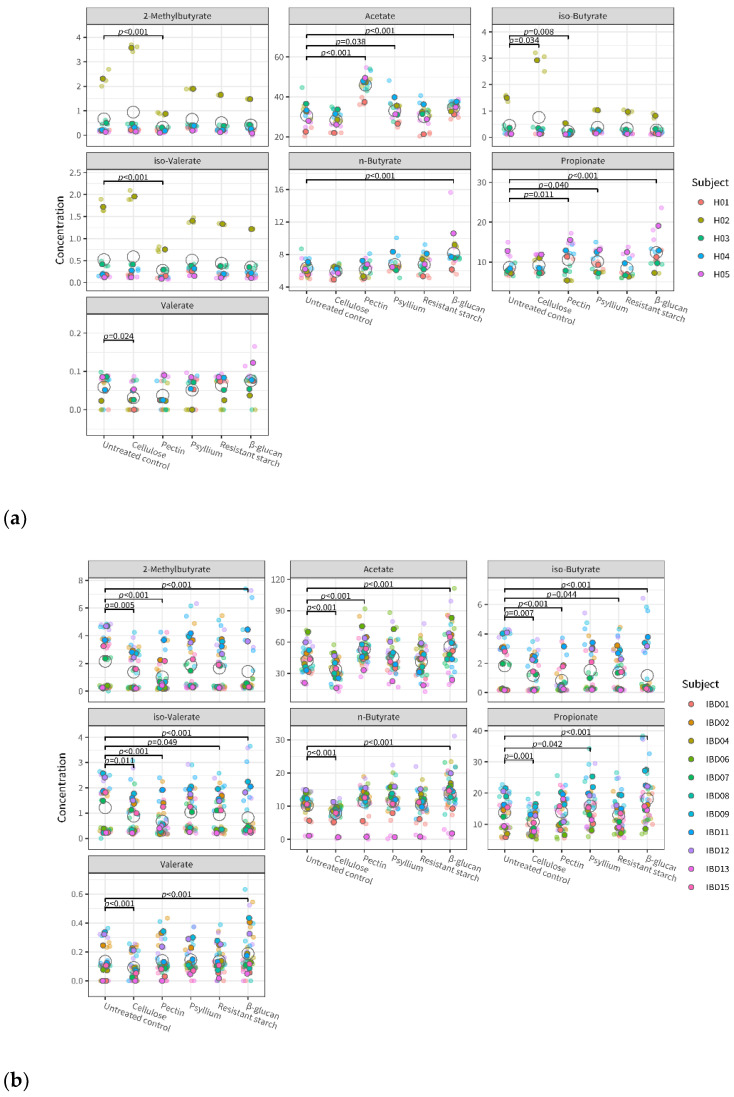
Metabolite levels in the microbiota of healthy (**a**) and IBD (**b**) subjects after fiber treatment and in untreated control conditions. Different colors indicate the individual donors, with smaller and more transparent dots indicating the individual replicates and larger and darker dots indicating the average of triplicates; open circles represent the average for each test condition. *p*-values indicate significant differences between fiber treatments and untreated control.

**Figure 6 microorganisms-11-02001-f006:**
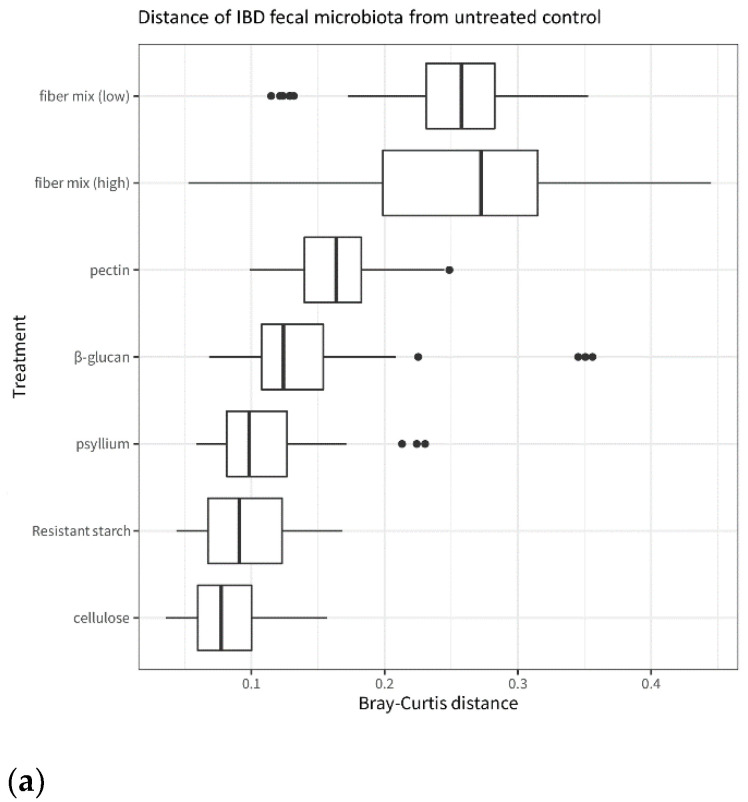
Distance from the untreated control of IBD microbiota treated with fiber and fiber mixes. (**a**): β-diversity of the IBD fecal microbiota after incubation with the different fibers and fiber mixes. The plot is based on the Bray–Curtis distances between each fiber treatment and the untreated control. (**b**): SCFA levels in the IBD fecal microbiota after incubation with the different fibers and fiber mixes. The plot is based on the Euclidean distances between each fiber treatment and the untreated control.

**Figure 7 microorganisms-11-02001-f007:**
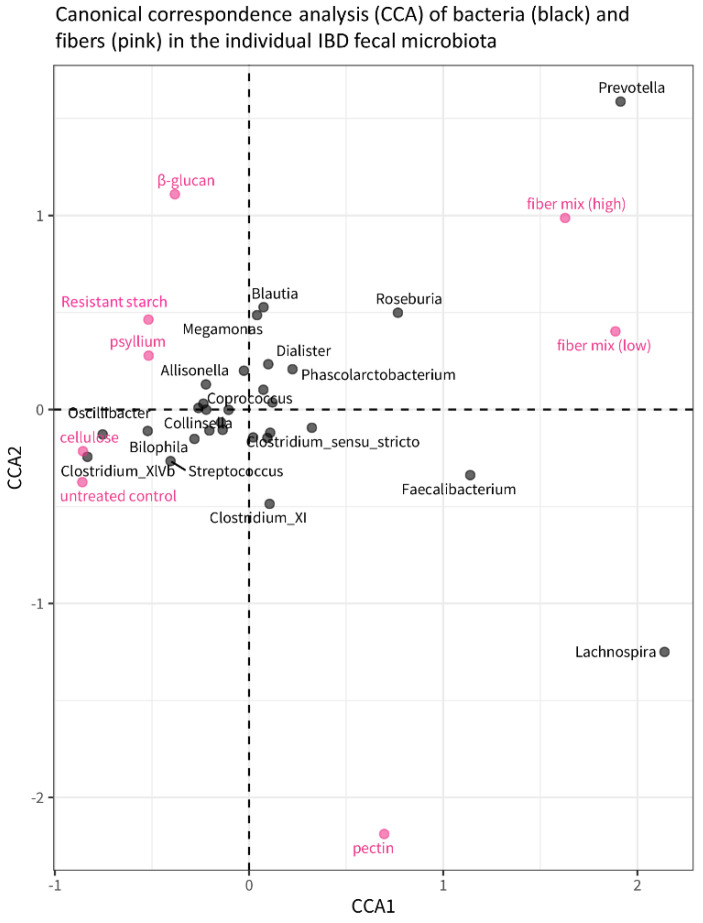
Canonical Correspondence Analysis (CCA) plot showing the relationship between fiber treatments and bacterial genera. The pink dots correspond to the different fiber treatments, and their position in the plot is determined by the microbiota composition that is, on average (across all study subjects), associated with those fibers. The distance between dots is, therefore, indicative of the differences in microbiota composition between different fiber treatments: dots/fibers that are closer to each other have a similar microbiota composition; dots/fibers that are further away from each other are more different in terms of microbiota composition.

## Data Availability

The sequencing data are available at the NCBI Sequence Read Archive under the BioProject ID PRJNA 914994.

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
