# Peer review of "Individual and Group-Based Effects of In Vitro Fiber Interventions on the Fecal Microbiota"

_microorganisms, 2023, doi:10.3390/microorganisms11082001_

Round 1

Reviewer 1 Report (Previous Reviewer 2)

General comments:

The manuscript is generally well written, with very few issues. Covers a good amount of novel research with good flow of the story.

Except for a few issues which I have stated below, I suggest this study to be accepted after minor corrections. 

Specific comments:

Line 77-78: In the statement, " The jar was kept cool until delivery at the laboratory 78 (within 24h from collection)." authors do not indicate the storage temperature. Were these samples stored at 4C, -20C or at dry ice? This is important, as storage temperature will determine all the downstream biochemistry from point of collection to the analysis.

Line 73: Please cite the ethics approval number.

Line 83 and elsewhere: When entioning any consumables or chemicals, please indicate (Catalog/model number, Company, City, State/province, Country) upon first mention. 

Line 90: By 'ion exchanged water' do authors meant "deionised water" or "MilliQ water"? If it is either of the two, simple words should be good enough to state it. 

Figure 4 and elsewhere: The comparison is between "non-IBD" and IBD individuals. When the term "Healthy" comes in, the definition of healthy needs must be clarified. This is important since the patients (as per the current manuscript's scope) are only healthy with respect to the IBD, but may have some other underlying issues (although they may be out of scope from this paper's scope). The article (https://www.ncbi.nlm.nih.gov/pmc/articles/PMC5778676/) would help the authors to differentiate. 

Author Response

Thank you for reading and reviewing our manuscript and for providing valuable suggestions. The following changes have been made based on your advice:

  • we indicated the storage temperature of the fecal samples, replacing the word "cool" which is too vague
  • we have added the ethical approval number, and also corrected the name of the METC committee that approved the study protocol and sample collection
  • we have added information (product number, CAS number, etc.) to the chemicals mentioned, and we also updated elsewhere in the methods section the information relative to other materials and instruments
  • we specified that Milli-Q water was used, rather than ion-exchanged
  • in the methods section we specified that "healthy", in the context of this paper, is limited to intestinal health. Although we agree that the subjects may have had other underlying conditions, they did not have any intestinal complaints (which many people might have without having received an official diagnosis for intestinal disorders). Thus, we decided to keep this descriptor, rather than changing it to "non-IBD" throughout the paper, given the scope of the paper and the focus on intestinal health. To emphasize this, we also, where appropriate, changed several instances of "health" to "intestinal health" in the introduction and discussion sections. For example, in the abstract we changed one instance of "healthy subjects" to "subjects with good intestinal health"

Reviewer 2 Report (Previous Reviewer 3)

The authors addressed most of my concerns.

Author Response

Thank you for reading and reviewing our manuscript: we are glad to hear that the concerns have been addressed, and we hope the manuscript is now acceptable for publication.

This manuscript is a resubmission of an earlier submission. The following is a list of the peer review reports and author responses from that submission.

Round 1

Reviewer 1 Report

This study describes the exposure of 16 individual fecal microbiota samples to 5 different fibers using an in vitro system for the anaerobic cultivation of bacteria. However, the author mentioned several limitations of this research, it cannot be ignored. There are major problems in the design of this study. First, the sample size of donors is too small and the baseline data is lacking. The age and sex of healthy people and IBD people were not matched. The above problems seriously affect the follow-up experimental results, and it is impossible to obtain accurate and convincing experimental data. In addition, several references are cited in the conclusion part, which should be explained in the discussion part. Second, the description of statistical methods is incomplete. Punctuation is used incorrectly. The use of references is not scientific and rigorous. In general, articles can not meet the requirements of magazine publication.

Author Response

Thank you for accepting to revise our manuscript. Here is a point-to-point response to the provided feedback

This study describes the exposure of 16 individual fecal microbiota samples to 5 different fibers using an in vitro system for the anaerobic cultivation of bacteria. However, the author mentioned several limitations of this research, it cannot be ignored. There are major problems in the design of this study. First, the sample size of donors is too small and the baseline data is lacking.
In the manuscript, we mention the small sample size and the lack of baseline data as limitations of this study. However, these limitations did not prevent us from addressing the research questions, especially considering the explorative nature of the study. One of the goals of this study was to detect interindividual and group differences in these effects – and we were able to detect both despite the sample size. Indeed, similar studies with even smaller sample sizes have also been able to demonstrate effects of compounds in in vitro microbial systems.
With regard to the baseline data, in our study we adopted a different approach, namely, we compared fiber-treated samples with untreated samples. The only consequence of a lack of baseline data is that it is not possible to make predictions of effects based on an initial situation. However, the goal of this study was not to make this type of prediction. For this reason, in the manuscript we don’t mention ‘responses’, we simply compare microbiota exposed to different test conditions. In the discussion, we indicate how the presence of baseline data would allow similar studies to make prediction. In short, we do not agree that the limitations of this study represent ‘major problems’.

The age and sex of healthy people and IBD people were not matched.
Matching is essential in human studies, where other biological factors associated with age and sex could confound the analyses. However, larger datasets are also required for proper matching. Given the type of this study (a preliminary, in vitro evaluation of fiber effects), and the sample size, matching would not be appropriate.

The above problems seriously affect the follow-up experimental results, and it is impossible to obtain accurate and convincing experimental data.
Although the study design presents some limitations, we disagree that these represent insurmountable problems, as explained in detail in our previous answers. In general, the data we presented, and the conclusions based on them, are accurate and convincing in light of the research questions and the methods applied to them. Furthermore, based on previous experience with the methods reported in the study, including the in vitro gut microbiota model, we have strong evidence that the data resulting from our experiment is accurate and reliable.

In addition, several references are cited in the conclusion part, which should be explained in the discussion part.
We have moved the references and the corresponding sentences to appropriate sections in the discussion

Second, the description of statistical methods is incomplete. Punctuation is used incorrectly. The use of references is not scientific and rigorous. In general, articles can not meet the requirements of magazine publication.
All parts of the manuscript have been previously revised for scientific soundness, clarity, language, style, and punctuation. To further improve our work, we would appreciate more specific feedback.

Reviewer 2 Report

General: I congratulate the authors for an excellent writeup of this article. While I was concerned for a low enrolment for this study, the authors have considered this a limitation, and use this as a preliminary test for a larger follow-up study. 

The storyline was good, easy to understand. The only issues are minor grammatical, formatting and unit errors (see below). Bit otherwise, I recommend this article to be accepted after minor corrections.

Specific:

Line 106: The mg/ml should be mg/mL. Same for Line 138

Line 138: ul should be replace with uL here and elsewhere4

Author Response

Thank you for accepting to revise our manuscript. We have applied the recommended changes throughout the manuscript.

Reviewer 3 Report

In this manuscript, Valeria Agamennone et al used an in vitro system for anaerobic cultivation of bacteria from 16 individual fecal microbiota samples, and found that the individual microbiota differed in composition and metabolite profiles after incubation with fibers. It is important and interesting to explore the effects of different fibers on microbial composition and functional profiles among individuals and between different health groups. In addition to the limitation including small sample size, lack of baseline information of microbiota composition before fiber fermentation, the authors did not well present their data.

1. It may be better to use the average of per sample that were tested in triplicate, rather than three values per sample.

2. In Figure 4, it was shown that very few bacteria were significantly increased by fibers, that may be due to that some individual lacked the bacteria species which could ferment fibers, that influence the statistical significant analysis. If each individual microbiota was compared and showed after different fiber treatment, it may be more clear to see the effect of these fiber on gut bacterial genera abundance.    

Author Response

Thank you for accepting to revise our manuscript and for your feedback. Here are our responses to the points raised:

With regard to point 2, you indicate that using the average of triplicates per sample is better than using three values per sample. The data analysis was performed following this line of reasoning, but I believe you might be referring to Figures 2, 3, and 5 (and to the Supplementary Figures) where the data points for individual triplicates are presented along with the average values (indicated by different transparencies). Although we agree with you for general visualization purposes, we also believe that reporting on the individual replicates is relevant in this case: as the manuscript presents results obtained from an in vitro model of the human gut microbiome, we think it is important to provide the reader with an idea of the technical reproducibility and variability between replicates that one can expect when using this technology. The figures thus purposefully display data points of the triplicates alongside their average to show the small dispersion of the individual replicates.

With regard to your second point, referring to the heatmap of figure 4: we would rather not alter this figure as it highlights the difference in the abundance of bacterial genera at the group level, but we do agree that this visualization does not provide any information on the individual microbiota. Therefore, we would like to add a supplementary figure which provides information on the changes at the level of bacterial taxa on an individual level (i.e. for each microbiota), instead of at a group level.

Round 2

Reviewer 1 Report

The authors have not made substantial changes in the manuscript, nor a reasonable explanation responded all my queries.  To ensure the scientificity and rigor of the magazine, I consider that it is not suitable for publication in Microorganisms.